# New Insights in Immunometabolism in Neonatal Monocytes and Macrophages in Health and Disease

**DOI:** 10.3390/ijms241814173

**Published:** 2023-09-16

**Authors:** Renske de Jong, Klaus Tenbrock, Kim Ohl

**Affiliations:** Department of Pediatrics, RWTH Aachen University, 52074 Aachen, Germany; redejong@ukaachen.de (R.d.J.); kohl@ukaachen.de (K.O.)

**Keywords:** neonatal tolerance, neonatal infection, bacterial, commensals

## Abstract

It is well established that the neonatal immune system is different from the adult immune system. A major task of the neonatal immune system is to bridge the achievement of tolerance towards harmless antigens and commensal bacteria while providing protection against pathogens. This is highly important because neonates are immunologically challenged directly after birth by a rigorous change from a semi-allogeneic sterile environment into a world rich with microbes. A so called disease tolerogenic state is typical for neonates and is anticipated to prevent immunopathological damage potentially at the cost of uncontrolled pathogen proliferation. As a consequence, neonates are more susceptible than adults to life-threatening infections. At the basis of a well-functioning immune response, both for adults and neonates, innate immune cells such as monocytes and monocyte-derived macrophages play an essential role. A well-responsive monocyte will alter its cellular metabolism to subsequently induce certain immune effector function, a process which is called immunometabolism. Immunometabolism has received extensive attention in the last decade; however, it has not been broadly studied in neonates. This review focuses on carbohydrate metabolism in monocytes and macrophages in neonates. We will exhibit pathways involving glycolysis, the tricarboxylic acid (TCA) cycle and oxidative phosphorylation and their role in shaping neonates’ immune systems to a favorable tolerogenic state. More insight into these pathways will elucidate potential treatments targets in life-threatening conditions including neonatal sepsis or expose potential targets which can be used to induce tolerance in conditions where tolerance is harmfully impaired such as in autoimmune diseases.

## 1. Introduction

### The Neonatal Immune System Is in a Tolerogenic State

Neonates (those <28 days of life) are born into a world rich in microbes as opposed to the semi-allogeneic sterile intra-uterine environment where fetal development takes place. This immediately results in a challenge to the neonate´s immune system as tolerance towards harmless antigens and commensal bacteria needs to be developed while protection against invading pathogens is also vital. The need to develop tolerance against commensal bacteria results in a higher susceptibility to infections, which has long been attributed to an immature innate immune system and lack of memory mechanisms of the adaptive immune system. However, numerous recent studies suggest that neonatal immunity is highly evolved, regulated and life stage appropriate. Not immaturity but a highly regulated tolerogenic status distinguishes the neonatal innate immune system from that of adults (reviewed by [1,2,3,4] among others). This disease tolerogenic state embraces the minimization of immunopathological damage, potentially at the cost of uncontrolled pathogen proliferation [5]. In contrast, the adult immune system uses a strategy that minimizes harm from pathogens (disease resistance) at the risk of host-inflicted damage. Comparing both strategies, disease tolerance and disease resistance, one major benefit of disease tolerance is that it is less energetically demanding [6]. For neonates, this is highly relevant considering the energetic constrains of early life due to limited energy reserves and the large demand of the newborn to sustain its core body temperature [7].

Worldwide, the leading causes of most neonatal deaths are preterm birth, intrapartum-related complications (neonatal asphyxia), birth defects and infections [8]. Hereby, it is important to keep in mind that there are major differences between countries/areas with higher or lower prosperity or poverty levels. In particular in middle- and lower-income countries, sepsis remains a leading cause of morbidity and mortality among neonates [9]. Neonatal sepsis is a life-threatening condition during the first 4 weeks of life commonly caused by bacteria such as *Staphylococcus aureus* (Gram-positive), *Enterobacter cloacae* (Gram-negative), *Klebsiella pneumoniae* (Gram-negative) and *Escherichia coli* (Gram-negative) [10,11]. Inappropriate control of infection can cause severe damage to different organs such as lung, kidney, liver and brain. Notably, premature born babies or babies that are born after birth complications are more prone to suffer from neonatal sepsis [12]. In addition, neonatal sepsis can result in long-term morbidity including neurologic sequelae, which is related to the particular vulnerability of the brain during this period. Neonates who experience sepsis are more likely to suffer from persistent (even for decades) neurodevelopmental alterations [13,14,15].

Although immune tolerance is a feature of the adaptive immune system, disease tolerance in neonates is regulated by the innate immune system. One of the key players to defend against infection, mutually for neonates and adults, is the macrophage. During embryonic development macrophages are produced and expanded by early progenitor cells and this takes place in the extraembryonic yolk sac [16]. Later, hematopoietic stem cells (HSCs) migrate to the fetal liver, which then serves as the major hematopoietic organ during embryonic development. Only in the perinatal period (depending on the definition: starts at 20th to 28th week of gestation and end 1 to 4 weeks after birth) bone marrow HSCs become the primary site of hematopoiesis [17]. Of interest, there are tissue-dependent differences in the ratio of presence of embryonically and postnatal-derived macrophages [18]. As an example, most alveolar macrophages are derived from embryonic progenitors and a smaller contribution is derived from HSCs. Embryonically derived fetal monocytes differentiate after birth into alveolar macrophages in a GM-CSF-dependent process and those macrophages maintain themselves without input of HSCs-derived blood monocytes [19,20]. It is unclear whether macrophages of distinct origin are functionally alike [18]. Generally, macrophages are essential for the elimination of invading pathogens (e.g., by phagocytosis and oxidative killing), playing an important role in immune modulation (e.g., by the secretion of pro-or anti-inflammatory cytokines and chemokines) as well as tissue homeostasis (e.g., by the secretion of resolution factors initiating tissue repair) [21,22]. In addition, macrophages function as antigen-presenting cells to induce adaptive immunity such as T cell activation [23]. To ensure that an optimal immune response is achieved (depending on situation and location) macrophages display high heterogeneity and plasticity.

Of fundamental importance for a well-responding macrophage, with its plastic and heterogenic character depending on environmental cues, is its cellular metabolism. Various cell intrinsic and extrinsic signals regulate the activity of metabolic pathways which are crucial for basic cellular maintenance and survival. Different cellular metabolic processes can initiate and alter immune effector functions [24,25]. This process, whereby intracellular metabolic pathways dictate immune functions, is called immunometabolism [26]. As an example, pro-inflammatory macrophages are more glycolytic, produce more reactive oxygen species (ROS) and show inhibited mitochondrial functions as measured by lower oxygen consumption rates [26,27].

Nowadays, immunometabolism has been widely studied in monocytes and macrophages from adults [28]. Without a doubt metabolic reprogramming is a prerequisite for heterogeneity and plasticity of adult macrophages. To date, only a few studies addressed metabolism in neonatal immune cells, although it is quite clear that metabolic circumstances and immune responses are significantly altered in neonates. Therefore, this review aims to describe the role of carbohydrate metabolism including glycolysis, the tricarboxylic acid (TCA cycle) and oxidative phosphorylation, which shape macrophage immune functions in neonates inducing tolerance against commensals, but protection against invading pathogens.

Understanding the role of immunometabolism at this critical age may have long-term therapeutic benefits for newborn children particularly when considering life-threating conditions such as sepsis. Additionally, understanding how neonatal tolerance against commensals is induced provide ideas how to improve current treatment methods regarding organs transplantations and autoimmune diseases, both examples of conditions whereby mechanism of tolerance fail.

## 2. Carbohydrate Metabolism

### 2.1. The Glycolytic Metabolic Pathway

Carbohydrate metabolism is a fundamental biochemical process that guarantees a constant supply of energy. In addition, the generation of additional metabolites is essential for function of the cell. The most vital carbohydrate is glucose, which is metabolized in a cascade of steps named glycolysis.

To start the process of glycolysis, exogenous glucose must be internalized by the cell. The uptake of glucose molecules occurs via glucose transporters (GLUT), which include to date 14 different isoforms and those transmembrane proteins are named GLUT1-14 (SLC2A1/14). GLUT1 is the most studied glucose transporter in immune cells. This transporter was found to be upregulated by activated (e.g., with liposaccharide (LPS) and/or INF-γ) inflammatory monocytes and macrophages to increase glucose [29]. Next to GLUT 1, monocytes express also profound levels of GLUT3, 6, 9 and 14 mRNA indicating that more than one isoform of the glucose transporters is involved in glucose uptake by monocytes [30]. For macrophages, GLUT 1, 3 and 5 appear to be the key transporters to take up glucose. Initial data showed a marked increase in GLUT 1, 3 and 5 in activated macrophages; however, those experiments were performed using THP-1 cells, a monocytic cell line derived from an acute monocytic leukemia patient [31]. It would be interesting to specify which isoforms are essential in glucose uptake in metabolic active primary macrophages both from adults and neonates.

After glucose uptake, glycolysis takes place in the cytosol. Two forms of glycolysis exist: aerobic and anaerobic glycolysis. Aerobic glycolysis, which takes place in the presence of oxygen, leads to the production of pyruvate. Subsequently, pyruvate is further processed by the mitochondrial TCA cycle (also known as Krebs cycle) followed by oxidative phosphorylation, altogether an efficient way to generate adenosine triphosphate (ATP). When oxygen is scarce, anaerobic glycolysis takes place in which pyruvate converts into lactate through lactic acid fermentation [32]. Although lactate itself is not utilized by the cell as a direct energy source, this reaction allows for the reduction of nicotinamide adenine dinucleotide (NAD)+ to NADH, which is used by numerous enzymes as a cofactor [26,32]. In addition, lactic acid fermentation ascertains the flow of glucose through glycolysis, generating two ATP per glucose molecule. Of note, anaerobic glycolysis provides ATP at a speed 100-fold faster than oxidative phosphorylation. Therefore, it is utilized to meet short-term energy goals of, for example, contractile skeletal muscles, but also rapidly proliferating cells such as immune cells [32]. In line with this, many pro-growth signaling pathways, including the phosphatidylinositol 3-kinase (PI3K) and mitogen-activated protein kinase (MAPK) pathways, promote cellular use of anaerobic glycolytic metabolism [26].

Next to providing ATP rapidly, glycolytic intermediates may be needed for the biosynthesis of certain molecules essential for proper polarization of immune cells to achieve an optimal immune response [2]. For instance, the glycolytic intermediate glucose-6-phosphate fuels the pentose phosphate pathway (PPP) in activated immune cells. Subsequently, this leads to the generation of precursors for nucleotides, amino acids and fatty acid synthesis which support cell proliferation and immune effector functions such as cytokine production [33]. In addition, the PPP provides NADPH, which is used for the production of ROS by NADPH oxidase [34]. Of note, ROS have antimicrobial properties as they induce oxidative stress in phagosomes from phagocytic cells such as macrophages leading to damage to bacterial cell structures and therefore impair the growth and survival of pathogens.

### 2.2. The TCA Cycle and Oxidative Phosphorylation

Following aerobic glycolysis, the TCA cycle is initiated. The biochemical reactions belonging to the TCA cycle occur in the matrix of mitochondria. This cycle, coupled to oxidative phosphorylation, generates ATP in a highly efficient manner and is used by cells whose primary requirement are energy and longevity (e.g., quiescent or non-proliferating cells). The TCA cycle is fueled by different sources, but mainly by glucose-derived pyruvate (residue of glycolysis), fatty acids and amino acids (e.g., glutamine). These nutrients are converted into acetyl coenzyme A (acetyl-CoA) which subsequently joins the TCA cycle by aldol condensation with oxaloacetate to form citrate. As well, glutamate is a critical fuel for the TCA cycle through its direct conversion into the TCA intermediate α-ketoglutarate. NADH and flavin adenine dinucleotide (FADH2) are the main products of the TCA cycle, which support oxidative phosphorylation by transfer of electrons to the electron transport chain.

In addition, the TCA cycle produces intermediate products for other purposes, e.g., for the production of amino acids and lipids needed for basic cell maintenance, but also to effectuate specific immune functions. For instance, in pro-inflammatory macrophages (also named classically activated macrophages or M1 macrophages, typically activated by LPS and/or INF-γ) the TCA cycle is interrupted in two places namely after citrate and after succinate [35,36]. The citrate molecules that accumulate in the mitochondria are exported via the citrate transporter and subsequently utilized for the production of fatty acids, which in turn are used for membrane biogenesis. Membrane biogenesis is essential for classically activated macrophages to sustain membrane production to support antigen presentation [37]. Succinate, the other metabolite that accumulates as a consequence of a broken TCA cycle, can inhibit prolyl hydroxylases leading to the stabilization of HIF-1α and the sustained production of pro-inflammatory cytokine IL-1β [35,36]. In addition, the metabolite itaconate is produced in high amounts in activated macrophages with an increased glycolytic flux [38]. Hereby, itaconate is made from cis-aconitate in the TCA cycle, upon LPS stimulation among other pro-inflammatory stimuli. Cis-aconitate, vacates the TCA cycle to be repurposed for itaconate production [39]. Most studies regarding itaconate functions point to a pronounced anti-inflammatory role [38]. One of these anti-inflammatory pathways comprises limiting succinate dehydrogenase (SDH) which subsequently blocks succinate-mediated inflammatory processes [40,41]. As well, itaconate induces anti-inflammatory and antioxidant signaling via nuclear factor erythroid 2 (NRF2) and activating transcription factor 3 (ATF3) [42,43,44].

On the other hand, alternatively activated macrophages (including M2 macrophages) have an intact TCA cycle coupled to oxidative phosphorylation [36]. Those macrophages are alternatively polarized by IL-4, IL-10 or IL-13 and promote anti-inflammatory pathways essential in wound healing and tissue repair [21,22]. An intact TCA cycle, among other things, allows the generation of UDP-GlcNAc intermediates that are necessary for the glycosylation of receptors expressed by alternatively polarized macrophages, such as the mannose receptor [33].

### 2.3. Differences in Carbohydrate Metabolism between Adults and Neonates

Numerous pathways have been described providing a deeper understanding of the role of metabolic reprogramming in the plasticity and function of macrophages (extensively reviewed elsewhere [33,45]); however, little is known about those pathways in neonatal macrophages. For instance, it has been long established in the adult immune system that to initiate inflammation, a shift from oxidative phosphorylation towards anaerobic glycolysis is required in order to provide energy and essential metabolic intermediates for the induction of immune modulatory pathways (similar to the Warburg effect in tumors) [46].

This shift, from oxidative phosphorylation towards anaerobic glycolysis, is severely diminished in monocytes from neonates born preterm resulting in low levels of pro-inflammatory mediators (e.g., INF-gamma and IL-6 release) in response to *Candida albicans* compared to monocytes from adults (Figure 1) [12]. This was corroborated by our group in macrophages, which were differentiated from cord blood monocytes and polarized toward a classical and alternative macrophage phenotype by INF-γ or IL-10, respectively. Here, a similar phenotype was observed with impairment of glycolysis [47]. Interestingly, reduced glycolysis in preterm monocytes and cord blood-derived macrophages did not impair phagocytosis, directing the need for glycolysis towards other immune effector functions [12,47,48,49,50].

On a molecular level, *MALT1* signalosome genes were found to be reduced in monocytes from neonates born preterm (Figure 1) [12]. In myeloid cells, the MALT1 signalosome is activated upon the recognition of fungal, bacterial and viral PAMPs by PRRs inducing inflammatory cytokine production including IL-6, IL-1β and TNF [51]. A loss of MALT1 function might explain, at least in part, an impaired response against *Candida* [12].

Besides an altered cytokine profile, neonatal murine peritoneal macrophages (<24 h) showed a diminished expression of the costimulatory molecules CD80 and CD86 compared to adults (42 days) (Figure 1). In line with this, their potential to induce T-cell proliferation upon LPS stimulation was significantly reduced, a response also observed in macrophages derived from cord blood monocytes [50,52].

In addition, on a genetic or epigenetic level, several metabolic regulators of glycolytic metabolism and the TCA cycle were found to be altered in CBMOs compared to PBMOs [47] (Figure 1). First, the expression of *GLUT1* was found to be reduced likely impairing the influx of glucose. Secondly, phosphofructokinase M (*PFKM*) mRNA was reduced, which is an enzyme needed to convert fructose 6-phosphate (F6P) into glyceraldehyde 3-phophate (GAP). As well, mRNA levels of mitochondrial pyruvate carrier 1 (*MPC1*) and 2 (*MPC2*), forming a heterodimer (MPC1/MPC2), were diminished. This heterodimer is essential to translocate pyruvate from the cytosol into the mitochondria. Moreover, the expression of the enzymes pyruvate carboxylase (*PC*) and malate dehydrogenase 2 (*MDH2*) were downregulated impairing the processing of pyruvate to oxaloacetate and malate to oxaloacetate, respectively. These molecular differences are in line with the absence of glycolysis induction upon immune stimulation observed in mononuclear phagocytic cells from neonates.

Regarding neonatal sepsis, a large screening approach, using genome-wide transcriptional analysis, was conducted to gain insight into immune-metabolic differences between healthy human neonates and neonates with bacterial infection [53]. In neonates, suffering from infection an increased expression of GLUT3 (SLC2A3) was observed. Glut3 is indeed the main glucose transporter in monocytes and can be activated upon stimulation [31]. There were no differences in Glut3 expression in macrophages suggesting a different regulation between monocytes and macrophages. A higher level of 6-phophofructo-2-kinase (PFK-2) that activates the glycolytic flux under hypoxic conditions and hexokinase 3 (HK3) that phosphorylates glucose to produce glucose-6-phosphate, the first rate-limiting step in glucose metabolism, were detected as well. How these metabolic differences are inducing a divergent immune response is an important research direction that needs to be explored.

Another metabolic regulator with a divergent expression between neonates and adults is the mammalian target of rapamycin (mTOR), a key regulator of the metabolic switch towards anaerobic glycolysis during immune activation [54]. mTOR transcripts were found to be elevated in cord blood compared to adult blood-derived macrophages, but total mTOR protein expression and mTOR phosphorylation are both reduced compared with adults [47]. This indicates post-transcriptional regulation of mTOR expression in neonates. In line with this, a reduced expression of 4EBP1, a downstream target of mTOR, was observed [12,47]. In addition, negative regulators of the mTOR pathway were upregulated namely NAD-dependent deacetylate sirtuin-1 (Sirt1) and DNA damage inducible transcript-4 like (Ddit4l) [12]. In adult mice, the mTOR pathway was found to be critical for survival of *Escherichia coli*-mediated sepsis [55]. Since mTOR signaling is diminished in neonates, these findings could explain a higher prevalence of sepsis and sepsis-mediated death in neonates.

Importantly, one study observed a rapid increase in glycolysis in cord blood macrophages (similar to adult macrophages) in response to LPS or *Mycobacterium tuberculosis* (Mtb) stimulation, the causative agent of tuberculosis [56]. It is possible that this observation, which deviates from the above-mentioned findings, can be explained by the fact that the macrophages used in this study where not polarized, were kept in culture for a longer time and a different pro-inflammatory stimuli was used.

### 2.4. External Factors Dictating Immunometabolism in Neonates

In pursuit of an altered glucose metabolism, in neonatal monocytes and macrophages, the following question arises: which environmental factors dictate a divergent macrophage metabolism in neonates? Or are they the result of genetically or epigenetically encoded developmental programs or a combination of both? 

The observed differences between neonatal and adult phagocytic cells can be partly explained by a concept called trained immunity [57]. This concept describes the long-term functional reprogramming (on average 3–12 months) of innate immune cells, which is induced by exogenous or endogenous stimuli and which leads to an enhanced response toward a second challenge after the return to a non-active state after a primary infection [58,59]. Hereby, epigenetic reprogramming of transcriptional pathways mediates trained immunity. Interestingly, trained immunity is accompanied by profound differences in a number of central metabolic pathways including glycolysis and oxidative phosphorylation [59,60].

As well, the counterpart of trained immunity, known as ‘innate immune tolerance’, may likely be involved in functional reprogramming of neonatal monocytes and macrophages. Hereby, immune cells are unable to activate gene transcription and its connected pro-inflammatory pathways upon re-stimulation by microbial particles [61]. As an example, recurrent exposure of macrophage to a high dose of LPS epigenetically enforced tolerance to prevent the expression of inflammatory genes [62].

Potential candidates, recently found to be involved in reprogramming neonatal phagocytic cells, are the S100 calcium binding proteins S100A8 and S100A9 and their extracellular heterodimer complex form S100A8/S100A9. These antimicrobial proteins were found in high amounts in human breast milk, fecal samples and serum from newborns [63,64]. S100 proteins, which bind Toll-like receptor (TLR) 4, regulate tolerance in neonatal immunity in several ways. As an example, breast milk-derived S100A8/A9, absorbed in the gut, stimulate lamina propria macrophages to support the expansion of regulatory T cells and in turn allow bacterial colonization [64]. Interestingly, PBMO treated with cord blood serum or S100A8/A9 suppressed p-mTOR activation and diminished glycolysis (Figure 2) [47]. These data lead to the hypothesis, that S100A8/A9 seem to diminish glycolysis pathways in neonatal macrophages to prevent a hyper-inflammation induced by commensal bacteria and potential pathogens that colonize the neonate in the first days after birth. In line with this, it has been shown that loss of S100A8/A9 resulted in a severe septic induced hyperinflammatory state in a model of neonatal *S. aureus* or LPS induced infection with a strongly enhanced mortality [63,65]. This severe septic state could be reversed by injecting neonatal pups with S100 proteins prior to sepsis induction [63]. Of interest, S100 proteins are massively released by newborn monocytes, induce immune responsiveness via MyD88 gene-dependent genes, prevent the excessive expansion of an inflammatory monocytes population, but did not impair defense against pathogens (Figure 2). So, S100 proteins ascertain effective dampening of the inflammatory responsiveness of innate immunity in newborns, but do not impair immune protection against infections [63,65]. As a consequence, a low expression of S100 proteins in neonatal fecal samples or serum particularly seen in preterm neonates was associated with the development of neonatal sepsis. Altogether, these data indicate a strong negative association between S100A8/A9 and sepsis risk in human newborns.

Next to S100 proteins, breast milk contains a great many of other biologically active components such as immunoglobulins, chemokines/cytokines, oligosaccharides, microRNAs, hormones, immune cells and microorganism all priming the perinatal immune system [66]. Of interest, a wide variety of compositional differences exist among individual mothers. While it has been shown that those components influence neonatal immunity, it has not been investigated whether and how these components affect metabolic pathways such as glycolysis, the TCA cycle and oxidative phosphorylation and thereby priming the neonatal immune system.

Furthermore, the detection of pathogens by innate immune cells dictates immune-metabolic pathways including macrophage metabolism and its associated polarization. Mechanistically, pathogen-associated molecular patterns (PAMPs), such as LPS, activate microbial sensors (also called pattern recognition receptors: PPRs) including C-type lectin receptors, nucleotide-binding oligomerization domain (NOD)-like receptors and TLRs. Of interest, signaling via PPRs lead to intracellular metabolic changes [67,68]. For instance, TLR4 activation by LPS leads to hypoxia-inducible factor-1α (HIF-1α) signaling and subsequently the upregulation of glycolytic pathways and downregulation of OXPHOS [69]. As a consequence, activated macrophages secrete a distinct set of cytokines (pro-inflammatory) to modulate inflammation [26]. Of interest, several pattern recognition receptors are differently expressed in neonates, children, adults and elderly [70], e.g., TLR2 expression on monocytes is lower in neonatal phagocytes compared to adults whereas no differences for TLR4 could be detected [71]. In addition, NOD1 expression was found to be decreased in preterm and term newborns compared with adults [72]. NOD2 was significantly lower in preterm babies, but not in term neonates, in comparison with adults. The capacity of monocytes to detect PAMPs in neonatal sepsis is crucial; however, it needs to be determined how PAMP-induced signaling is altered in neonatal monocytes and how metabolism plays a role in this.

Differently programmed neonatal mononuclear phagocytic cells may also be the result of epigenetic changes and transcriptional remodeling. Epigenetic mechanisms can activate or silence gene transcription by modulating chromatin structure and stability without affecting the DNA sequence itself. Epigenetic modifications include DNA methylation, histone modifications and microRNA (miRNA) expression. These modifications are known to regulate immune cell differentiation and function in adults but have sparsely been studied in neonates. In neonatal monocytes a unique histone modification landscape was observed [73]. During development from neonates to adults, monocytes lose the poised enhancer mark H3K4me1 and gain the activating mark H3K4me3. The addition of three methyl groups (me3) on lysine (K) 4 of histone (H) 3 (i.e., H3K4me3) leads to activation of gene transcription. In contrast, H3K4me1 silences transcription [74,75]. This indicates that promoters in preterm neonates are less open and accessible to transcription factors compared to term neonates and this development continues in the following months (Figure 2). Of note, the abundance of the poised enhancer mark H3K4me1 and the activating mark H2K4me3 at immunologically important neonatal monocyte gene promoters, including *IL-1β*, *IL6*, *TNF* and *CCR2*, was associated with a matching gene expression [73].

Additionally, in human monocytes from neonatal cord and adult blood, a difference in miRNA-146 was observed. MiRNAs are small, single-stranded, non-coding RNA molecules (21 to 23 nucleotides) and are involved in RNA silencing and post-transcriptional regulation of gene expression. More specifically, miRNA-146 was found to be a regulator of Toll-like receptor 4 (TLR4) through a negative feedback loop mechanism [76]. MiRNA-146 (both miRNA-146a and b) showed a time-dependent upregulation which suggest a role for this miRNA in the development and function of the innate immune system of neonates (Figure 2) [77]. To what extent epigenetic modifications, such as histone modifications and miRNA expression, could explain the observed differences between neonatal and adult phagocytic mononuclear cells regarding carbohydrate metabolism has not yet been investigated to date.

Interestingly, trained immunity and effective protection against infection can be inherited from parents [78]. A recent study showed that the offspring of trained mice exhibit transcriptional and epigenetic changes and showed improved protection against *Escherichia coli* and *Listeria monocytogenes*. Hence, transgenerational immune modifications could be beneficial to survive dangerous pathological conditions and may limit innate immune responsiveness during bacterial colonialization in early fetal life.

In conclusion, an altered glucose metabolism and its related immune modifications, observed in neonates, are the result of environmental factors (e.g., S100A8/9 in breast milk) and epigenetic changes that may partly be inherited from parents.

## 3. Conclusions and Future Perspectives

Emerging evidence show significant differences between neonatal and adult phagocytic cells regarding immunometabolism. One recurring difference is that neonatal monocytes and macrophages exhibit lower rates of aerobic glycolysis due to epigenetic alternations connected to a life stage compatible microenvironment. The myth that the neonatal immune and metabolic system is an underdeveloped version of the adult system does not match the published findings. Currently, it is hypothesized that the observed dissimilarities, between newborns and adults, are attributed to the fact that both groups have different health benefiting priorities considering their developmental state. The adult immune system is primed to remove the invading pathogens at the cost of tissue injury (disease resistance). The fetal immune system is more tolerant towards invading microbes (disease tolerance) as allowance is needed for commensal bacteria to colonize the gut. Of note, when microbial colonialization starts is not precisely known. Novel findings indicate that prior to birth, already certain bacterial and fungal strains are present in the womb determining immune cell fate and functions of the neonate [79,80,81]. However, evaluation of these studies indicates that the detected microbial signals are likely the result of contamination during the clinical procedures to obtain fetal samples or during DNA extraction and DNA sequencing [82]. Nonetheless, this disease-tolerant state, observed during ontogeny, is less energy demanding and potentially less harmful, which is highly relevant as newborns have limitations in energy supply and use much of their energy to grow.

The critical balance, which seems to exist between disease resistance and disease tolerance, shifts between different life stages. In addition, disturbance of this balance may lead to fatal consequences, such as uncontrolled microbial infections. In a healthy state, the neonate is better off in a disease-tolerant state as this allows for antigen sensitization and colonialization by harmless pathogens in the first days and weeks after birth. However, under inflammatory conditions, such as sepsis, a more disease-resistant state is desirable. This potentially explains a higher incidence of infection in preterm and term newborns. Interestingly, septic neonates can bear a much higher number of bacteria in blood compared to septic adults [5,83,84]. One question that arises is ´if´ and ´how´ this balance can be targeted in pathophysiologic conditions that occur in neonates. Hereby, more insight into neonatal immunometabolism is crucial.

Of interest, adult patients with severe sepsis that result in immunoparalysis show a phenotype with similarities to the neonatal immune metabolic phenotype. During immunoparalysis, leukocytes show broad metabolic defects including lower rates of aerobic glycolysis [55]. Notably, it has been demonstrated that restoring the glycolytic capacity of leukocytes by IFN-γ leads to an increased cytokine production benefiting survival rates of septic mice [55]. In addition, re-education of immune cells is also considered as pharmacotherapy in cancer. Hereby, immunosuppressive tumor-resident cells, which also show reduced levels of aerobic glycolysis and are linked to a poor disease prognosis, are primed towards pro-tumorigenic activities whereby the metabolic state of the leukocytes is important [85,86]. As an example, one group found that agonistic anti-CD40 treatment led to metabolic reprogramming of tumor-associated macrophages. The treatment with anti-CD40 rewired fatty acid and glutamine metabolism inducing anti-tumorigenic polarization of the macrophages [87].

Before applying immunomodulatory interventions in newborns suffering from sepsis as described above, a deeper understanding of neonatal immunometabolism at cellular and molecular level is needed. One major limiting factor here is the amount of suitable research material. Most research is performed using cord blood as this is commonly the major reliant source of immune cells from newborns. The umbilical cord is a waste product and the use of this source for immune cells does not harm the baby or the mother. Herewith, it is important to keep in mind that knowledge obtained by using cord blood cannot be translated one-to-one into postnatal immunity as immunological alterations, driven by microbial interactions, occur rapidly after birth [88]. To overcome this problem, other mammalian species could be used such as mice. However, hereby caution is needed as differences regarding immunometabolism between mice and men are only starting to be mapped properly and substantial differences have been observed [89,90].

In addition, knowledge obtained using cord blood cannot always be translated to immune cells residing at different immune sites. A tissue/organ-specific immune pathway is, for example, observed in the peritoneal cavity of neonatal mice. In mouse peritoneal neonatal macrophages, an altered cytokine and chemokine profile was observed whereby the pro-inflammatory factors MIP-1α, MCP-1 and IL-6 were upregulated compared to peritoneal macrophages from adult mice [91]. The upregulation of MIP-1α and MCP-1 points towards a high chemoattractive potency of neonatal murine macrophages, which fits to an increased influx of neutrophils into the peritoneal cavity (upon LPS stimulation) which was subsequently observed [91]. Moreover, another research group showed an impaired ability of alveolar macrophages from infants (age range from 6-23 months) to control *Mycobacterium tuberculosis* [92]. Hereby, a different functionality of immunometabolic pathways might explain the differences. In addition, in adult human alveolar macrophages, the metabolic shift from glycolysis towards oxidative phosphorylation upon LPS stimulation was not observed as for cord blood-derived macrophages. This indicates that human alveolar macrophages, in comparison to cord blood-derived macrophages, show a different metabolic phenotype and that alveolar macrophages rely more on oxidative phosphorylation [93,94]. One possible cause for the observed differences might be that nutrients are not equally available to all immune cells and that immune cells adjust their responses accordingly [95].

Although time is needed to develop immunometabolic modulatory interventions to treat neonatal sepsis, knowledge on immune-metabolic pathways are starting to be used as a diagnostic tool to detect and predict severity of infections [96]. In relation to this, two studies showed that by using urine samples, early- and late-onset neonatal sepsis can be predicted [97]. Using both nuclear magnetic resonance (NMR) spectroscopy and liquid or gas chromatography–mass spectrometric (LC–MS or GC–MS) analysis, a distinct metabolic profile could be identified between term- and preterm septic neonates and controls. The metabolic profile of septic neonates is characterized by an increased amount of glucose, pyruvate, lactate and acetate besides lower levels of glutamine, vitamin of B (both B2 and B3) and pentose phosphate pathway metabolites [97,98].

A greater understanding is required regarding the link between metabolism and immunity in neonates. Not only our knowledge regarding the link between carbohydrate metabolism and immunity needs to be expanded, but also other metabolic pathways, such as lipid metabolism (e.g., fatty acid oxidation and fatty acid synthesis), nucleotide metabolism and amino acid metabolism, are of particular interest and sparsely studied in neonatal immunity. This knowledge may be used to fine-tune the balance between disease resistance and tolerance benefiting neonates suffering from infections and other immune-related pathologies.

## Figures and Tables

**Figure 1 ijms-24-14173-f001:**
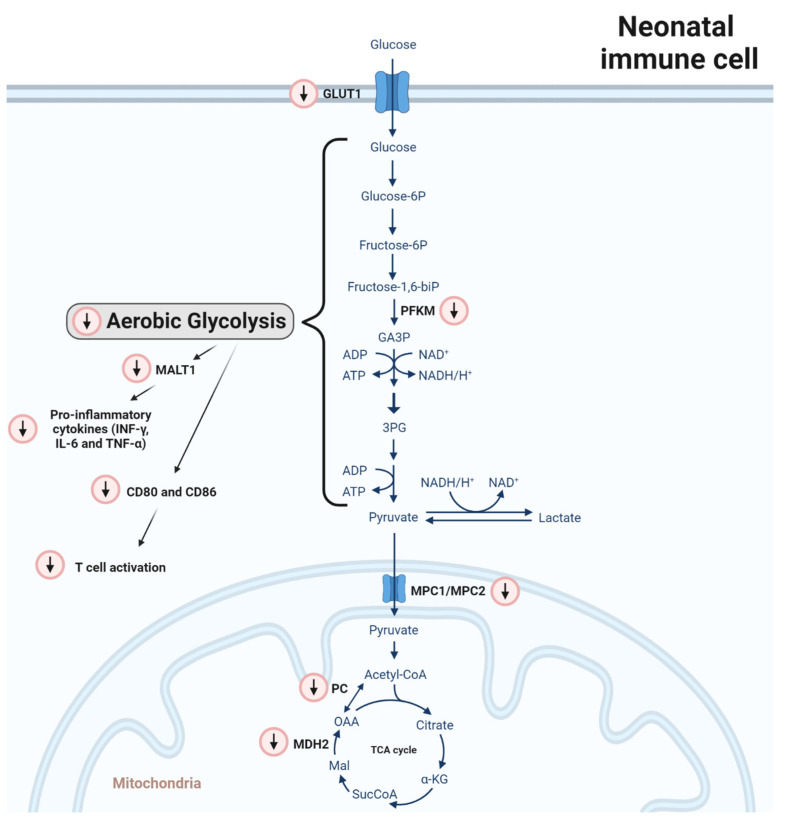
Differences in carbohydrate metabolism between adult and neonatal monocytes/macrophages. Red-circled arrows indicate immunometabolic differences in neonates in relation to adults. In neonatal mononuclear phagocytic cells (monocytes and macrophages), *GLUT1* and phosphofructokinase M (*PFKM*) mRNA are reduced. In line with this, a consecutive reduction in glycolysis activity leads to a lower expression of *MALT1* signalosome genes, which are induced after pattern recognition receptors (PAMPS) activation. Consequently, pro-inflammatory cytokine production in neonates is reduced and the expression of the costimulatory molecules CD80 and CD86 is abrogated leading to impaired induction of T cell proliferation upon LPS stimulation. Regarding the TCA cycle, *MPC1* and *MPC2* gene expression is reduced. The MPC1/MPC2 heterodimer translocates pyruvate from the cytosol to the mitochondria. In addition, the gene expression of the enzymes pyruvate carboxylase (*PC*) and malate dehydrogenase 2 (*MDH2*) is downregulated, impairing the processing of pyruvate to oxaloacetate and malate to oxaloacetate, respectively.

**Figure 2 ijms-24-14173-f002:**
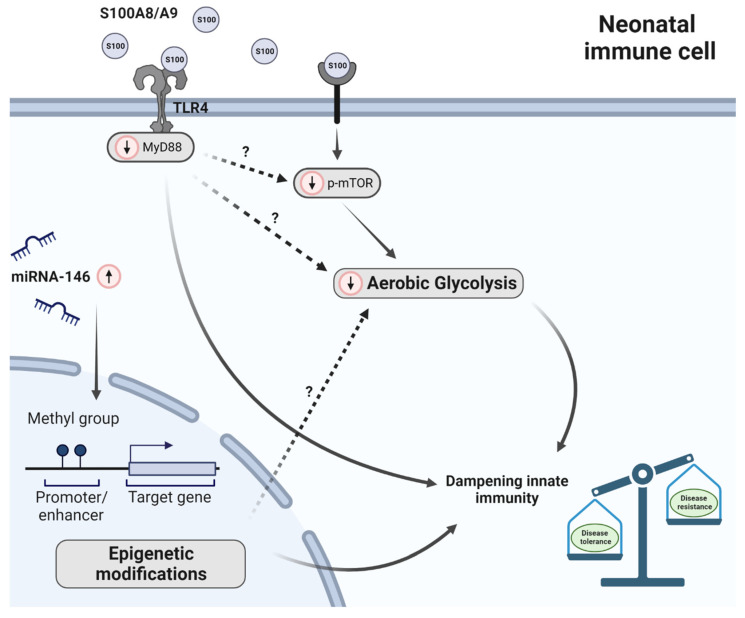
Factors dictating immunometabolism in neonates. Red-circled arrows indicate immunometabolic differences in neonates in relation to adults. In breast milk, fecal samples and serum of neonates the antimicrobial proteins S100A8 and S100A9 (forming a heterodimer) are upregulated. S100 proteins diminish glycolysis and subsequently inflammation induced by pathogens. This is (at least partly) regulated by mTOR. In neonates, mTOR phosphorylation (p-mTOR) was reduced. Subsequently, suppression of p-mTOR activation leads to reduced glycolysis activity. In addition, S100A8 and S100A9 induce immune responsiveness via MyD88-dependent genes. Furthermore, epigenetic alternations were observed in preterm neonates compared to term neonates; however, it has not been studied to date how carbohydrate metabolism is affected by this. Unsolved questions with respect to glycolysis are depicted with a question mark. Furthermore, miRNA-146 (both miRNA-146a and b) is upregulated in human neonatal monocytes. Altogether, these differences lead to an immune system that is primed to induce disease tolerance instead of disease resistance.

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
