# Peer review of "New Insights in Immunometabolism in Neonatal Monocytes and Macrophages in Health and Disease"

_ijms, 2023, doi:10.3390/ijms241814173_

Round 1

Reviewer 1 Report

The manuscript by de Jong et al provides an interesting, well organized description of the differences in immunometabolism within monocytes and macrophages between neonates and adults.  They emphasize the different strategies of infectious disease tolerance and disease resistance and why each is appropriate for the age group but with its consequences.  Other than some typos and clarifications (see below), this manuscript is acceptable.

Specific Comments

Please review manuscript for proper use of plural and singular, and change vs difference.

36.  Is lack of memory due to mechanisms or library of memory cells.  If only innate is being considered, then what mechanisms are being considered?  Trained immunity?

77 Replace impaired  with inhibited ?

83 typo: replace ‘know’ with now

138 replace supporters with support

147 rephrase as: generates ATP in a highly efficient manner and is used….

151 replace with   subsequently

154 format for FADH2

171 typo: regarding

191 typo:  a classical…..

251 replace ‘changes’ with ‘differencs’

256 please use a better term than ‘agonized’

316 is ‘regulations’ the intended term?

351 typo:  may partly…..

354 replace changes with differences

357 rephrase as: mTOR phosphorylation was reduced (p-mTOR).

359 typo  dependent

384 replace with     colonized

384 typo  bear

401 rephrase as ‘the treatment with anti-CD40…..’ 

412  replace ‘bird’ with birth

414 replace ‘started’ with starting

430 Delete ‘Hence’

431 rephrase: ‘Not only is our knowledge regarding carbohydrate metabolism limited, but…….

Very interesting and well written paper with typos that need to be corrected.  Noted in specific comments. 

Author Response

Please review manuscript for proper use of plural and singular, and change vs difference.

  1. Is lack of memory due to mechanisms or library of memory cells.  If only innate is being considered, then what mechanisms are being considered?  Trained immunity?

Thanks for the hint. This is indeed related to adaptive immunity and clarified now.

77 Replace impaired  with inhibited ?

83 typo: replace ‘know’ with now

138 replace supporters with support

147 rephrase as: generates ATP in a highly efficient manner and is used….

151 replace with   subsequently

154 format for FADH2

171 typo: regarding

191 typo:  a classical…..

251 replace ‘changes’ with ‘differences’

256 please use a better term than ‘agonized’

316 is ‘regulations’ the intended term?

351 typo:  may partly…..

354 replace changes with differences

357 rephrase as: mTOR phosphorylation was reduced (p-mTOR).

359 typo  dependent

384 replace with     colonized

384 typo  bear

401 rephrase as ‘the treatment with anti-CD40…..’ 

412  replace ‘bird’ with birth

414 replace ‘started’ with starting

430 Delete ‘Hence’

431 rephrase: ‘Not only is our knowledge regarding carbohydrate metabolism limited, but…….

Dear reviewer 1,
thank you for your kind word. Your feedback was very clear and helpful. We made adjustments to the text accordingly.

Reviewer 2 Report

The need to understand how the neonatal immune response differs from that of adults is driven by the mortality and morbidity linked to infection and sepsis in newborns and infants, is the motivation behind the current work entitled, "New Insights in Immunometabolism in Neonatal Monocytes and Macrophages in Health and Disease," which summarized the research findings that support the role of cellular metabolism in determining immune cell fate and functions. The review has potential but have several loopholes. There are recommendations mentioned below:

Comments:

1.      To challenge the dogma that human fetus is sterile is difficult. It had been long debated by perinatal and microbiological researchers as to whether the human microbiome is seeded prior to birth or not. In a healthy pregnancy the fetus is considered a sterile environment devoid of any living microorganisms under physiological conditions. However, there are findings which suggests that microbial colonization or mutualistic symbiosis is considered to occur during and following deliver. Comment?

2.     Neonatal mortality incidence reports need to be mentioned. The authors must write in the introduction section about the major causes of neonatal deaths in addition to sepsis and later highlight sepsis broadly. Also, world-wise/geographical death rates and diagnosis aspects can be emphasized in neonatal septicemia.

3.     The pathophysiology of sepsis is largely from investigations in adult populations, including both humans and animals but neonatal is important to understand. Even in comparison with children, neonates manifest a unique host immune response to septic shock. Kindly add data on this.

4.     The authors could describe about postnatal neonatal sepsis as systemic inflammatory response syndrome (SIRS) in the introduction section.

5.     Neonatal macrophages origin must be mentioned.  The phenotypic expression levels of macrophages in the neonatal lung and their functional significance in the pulmonary immune defense during the infections need to be written. The authors can also highlight the impaired macrophages and their implications in pathogenesis of specific neonatal diseases.

6.     The authors have highlighted the ROS mediated pathways of macrophages. What about RNS?

7.     Aspects on metabolic reprogramming to macrophage plasticity and function is not highlighted well.

8.    Role of TLRs is not shown? The authors must elaborate on TLRs expression on neonatal monocytes/macrophages in response to microbes.

9.     The authors nowhere discussed about the different macrophage activation types, (M1 and M2). Their reprogramming is dependent on metabolic changes to promote phenotypic switching. Seems like a missed opportunity.

10.There are reports on metabolic reprogramming alterations induced by Toll-like receptors (TLRs), and metabolism mediators like NOD-like receptors (NLRs) in neonates. The authors must highlight it.

11.  There are studies suggesting microbiome affects macrophage development and polarization, granulocyte numbers and hematopoiesis during early life. Add it?

12. In section, 2.1 and 2.2, the authors have described in detail about the glucose metabolism via glycolysis, TCA, and others. While these are universal processes and don’t need much explanations in depth. The authors must target the current problem as it seems the topic is deviated.

13. What microbiota are present in neonates in general? What specific type of bacteria are involved or largely seen in Neonatal Sepsis (Gram+ or gram-ve?)? The presence of lactobacillus and its links to sepsis and activation of macrophages is not shown.

14. In addition to the Glut3, there are reduced expression GLUT1 observed in cord blood-derived macrophages.  GLUT1 is the primary rate-limiting glucose transporter on proinflammatory-polarized Macrophages. The authors must show such recent studies.

15. There is a need to describe about pregnant women and infants.  Maternal immune status during pregnancy also appears to affect neonatal and infant immune development. What differences are observed in infants as a result of maternal metabolic syndromes and chronic inflammatory conditions during infections?  Kindly add.

16. The post-natal immunometabolism aspects are untouched. They can be briefly written.

17. An altered immune cell compositions especially of macrophages is seen in neonatal intestines with alterations in HIF1-a. HIF-1α is a critical hub that integrates hypoxic and immunogenic signals during infection or inflammation. Also it regulates the expression of genes encoding for glycolytic enzymes. Did the authors found studies in similar lines?

18.  Are there any reports on use of probiotics in preterm neonates for anticipation of mortality and late onset sepsis? Kindly mention.

19. The authors have briefly described the breast milk. While there are studies suggesting that monocytes increased the ability to differentiate in GMSF and the immune components of human milk can be changed during an infection in the nursing infants. Kindly add it.

20.                       Both figures have issues. Figure one is more entitled to show limited factors responsible for comparatively changing in adults and neonates. While more emphasis is shown on universal methods. The figure 2 is explaining more about the two heterodimers S100A8 and S100A9 which is fine as neonates show highly elevated blood levels of S100A8/A9 alarmins. A wholistic approach about the pathomechanism is missing.

21. Clinical aspects and significance are missing.

22. Minor grammar issues: In line 83, “Up to know” should be written as up to now. In line 135, “immune response2” Please correct such mistakes.

The manuscript is well written with perfect English language. There are some grammatical issues which can be modified.

Author Response

Reviewer 2:  

The need to understand how the neonatal immune response differs from that of adults is driven by the mortality and morbidity linked to infection and sepsis in newborns and infants, is the motivation behind the current work entitled, "New Insights in Immunometabolism in Neonatal Monocytes and Macrophages in Health and Disease," which summarized the research findings that support the role of cellular metabolism in determining immune cell fate and functions. The review has potential but have several loopholes. There are recommendations mentioned below:

Dear reviewer 2,
thank you for your clear and helpful feedback. We tried to incorporate all points into the manuscript without losing the focus of our review. In our opinion, some substantive points are beyond the scope of this review and where therefore only mentioned briefly in the manuscript. We hope you agree that our review has visibly improved.

Comments:

  1. To challenge the dogma that human fetus is sterile is difficult. It had been long debated by perinatal and microbiological researchers as to whether the human microbiome is seeded prior to birth or not. In a healthy pregnancy the fetus is considered a sterile environment devoid of any living microorganisms under physiological conditions. However, there are findings which suggests that microbial colonization or mutualistic symbiosis is considered to occur during and following deliver. Comment?

This is a subject of debate. There are recent studies suggesting that the human fetus and the prenatal intrauterine environment  are stably colonized by microbial communities in a healthy pregnancy, however the detected microbial signals are likely the result of contamination during the clinical procedures to obtain fetal samples or during DNA extraction and DNA sequencing (PMID: 36697862)..  To better reflect that there is an ongoing discussion about this topic, we added some references (ref. 78, 79 and 80, PMID: 36697862) on this topic in the discussion section (line 405 ff.), and discuss this in this section.

  1. Neonatal mortality incidence reports need to be mentioned. The authors must write in the introduction section about the major causes of neonatal deaths in addition to sepsis and later highlight sepsis broadly. Also, world-wise/geographical death rates and diagnosis aspects can be emphasized in neonatal septicemia.

In the introduction section we included the major causes of neonatal deaths. Subsequently, we introduced neonatal sepsis (line 49 ff). We shortly highlighted that neonatal sepsis mainly occurs in middle and lower-income counties and that there are major differences between counties/areas. We did not include geographical data in detail, because we think this is beyond the scope of this review.

  1. The pathophysiology of sepsis is largely from investigations in adult populations, including both humans and animals but neonatal is important to understand. Even in comparison with children, neonates manifest a unique host immune response to septic shock. Kindly add data on this.

We thank the reviewer for his/her valuable suggestion and can only agree that comparing neonates with children would be valuable. However, we feel that this topic should be better suited to be discussed in a separate manuscript which focus on the child´s development from neonate to adult.

  1. The authors could describe about postnatal neonatal sepsis as systemic inflammatory response syndrome (SIRS)in the introduction section.

It is true that neonatal sepsis can be described as a systemic inflammatory response syndrome. However, we believe that including this term in the manuscript does not benefit the readability of the manuscript and that it is not necessary to convey its core message.

  1. Neonatal macrophages origin must be mentioned.  The phenotypic expression levels of macrophages in the neonatal lung and their functional significance in the pulmonary immune defense during the infections need to be written. The authors can also highlight the impaired macrophages and their implications in pathogenesis of specific neonatal diseases.

In the introduction section we included the origin of neonatal macrophages (ref. 15, 16 and 17) (line 66 ff). We also highlighted that the origin of macrophages differ between tissues. Hereby we focused on alveolar macrophages in the lung (ref. 18 and 19). We did not include much detail on the functional significance of those differences as this is still largely being investigated.

We could not find relevant papers which link macrophage origin and immunometabolism in neonates. We could find this for alveolar macrophages in adults and we included those references (ref. 92 and 93)  in the discussion (line 462 ff).

  1. The authors have highlighted the ROS mediated pathways of macrophages. What about RNS?

We agree with the reviewer that RNS mediated pathways of macrophages are interesting, however, macrophages execute many effector pathways (e.g. lipid mediators, amino acids etc.). In this manuscript we used ROS as an example of an effector pathways executed by macrophages kill microbes.

  1. Aspects on metabolic reprogramming to macrophage plasticity and function is not highlighted well.

We believe focussing on metabolic reprogramming  and its role in macrophage plasticity and function is beyond the scope of the current review. In addition, there are several good reviews on this topic. In the new version of the manuscript we included two more references (ref. 32 and 44) to direct our readers towards those articles (line 196 ff.).

  1. Role of TLRs is not shown? The authors must elaborate on TLRs expression on neonatal monocytes/macrophages in response to microbes.

We agree with the reviewer that it is important to elaborate on TLRs and NLRs (see point 10) expression in neonatal monocyte/macrophage response in microbes. We included an entire paragraph on pattern recognition receptors and its link with immune-metabolic pathways in neonates. To highlight the importance of TLRs and NLRs we included the following references: 66, 67, 68, 69, 70 and 71 (line 326 ff.).

  1. The authors nowhere discussed about the different macrophage activation types, (M1 and M2). Their reprogramming is dependent on metabolic changes to promote phenotypic switching. Seems like a missed opportunity.

We described the macrophage phenotypes as classical and alternatively activated macrophages which can be respectively named as M1 and M2 macrophages. We included this way of naming to the manuscript.   

  1. There are reports on metabolic reprogramming alterations induced by Toll-like receptors (TLRs), and metabolism mediators like NOD-like receptors (NLRs)in neonates. The authors must highlight it.

See point 8.

  1. There are studies suggesting microbiome affects macrophage development and polarization, granulocyte numbers and hematopoiesis during early life. Add it?

This is  beyond the scope of the review. Microbiotic influences have already been revised elsewhere.

  1. In section, 2.1 and 2.2, the authors have described in detail about the glucose metabolism via glycolysis, TCA, and others. While these are universal processes and don’t need much explanations in depth. The authors must target the current problem as it seems the topic is deviated.

In our opinion those sections contribute to the readability of the manuscript. We have the experience that physicians and scientist, especially working in different research fields which are not related to immunometabolism, do not always have the knowledge regarding those basic biological processes by hand. 

  1. What microbiota are present in neonates in general? What specific type of bacteria are involved or largely seen in Neonatal Sepsis (Gram+ or gram-ve?)? The presence of lactobacillus and its links to sepsis and activation of macrophages is not shown.

We included reference 9 and 10 to answer this review question (line 58). We did not include the link between lactobacillus, sepsis and the activation of macrophages as the focus of the manuscript is on immune-metabolic pathways in monocytes/macrophages and not on macrophage activation in general.

  1. In addition to the Glut3, there are reduced expression GLUT1 observed in cord blood-derived macrophages.  GLUT1 is the primary rate-limiting glucose transporter on proinflammatory-polarized Macrophages.The authors must show such recent studies.

We included reference 28 in which GLUT1 is described as the primary rate-limiting glucose transporter on pro-inflammatory macrophages (line 120 f).

  1. There is a need to describe about pregnant women and infants.  Maternal immune status during pregnancy also appears to affect neonatal and infant immune development. What differences are observed in infants as a result of maternal metabolic syndromes and chronic inflammatory conditions during infections?  Kindly add.

We agree with the reviewer that it is highly interesting how chronic metabolic syndromes influences the disease process of neonatal sepsis. In our opinion, this subject is better suited in a review with the focus on this theme specifically.

  1. The post-natal immunometabolism aspects are untouched. They can be briefly written.

In accordance with point 3, we think that post-natal immune-metabolic aspects are better suited to be discussed in a separate manuscript which focus on the child´s development from neonate to adult.

  1. An altered immune cell compositions especially of macrophages is seen in neonatal intestines with alterations in HIF1-a. HIF-1α is a critical hub that integrates hypoxic and immunogenic signals during infection or inflammation. Also it regulates the expression of genes encoding for glycolytic enzymes. Did the authors found studies in similar lines?

As we mostly focus on blood derived macrophages, the intestinal immune cell composition is not a topic of our review.

  1. Are there any reports on use of probiotics in preterm neonates for anticipation of mortality and late onset sepsis? Kindly mention.

Yes, those reports exists, however, we decided not to include them as our manuscript is focusing on immunometabolism and we did not found a published paper on probiotics, neonates and immunometabolism (focus on carbohydrate metabolism).

  1. The authors have briefly described the breast milk. While there are studies suggesting that monocytes increased the ability todifferentiate in GMSF and the immune components of human milk can be changed during an infection in the nursing infants. Kindly add it.

Also these interesting studies would be the topic of a different review.

  1. Both figures have issues. Figure one is more entitled to show limited factors responsible for comparatively changing in adults and neonates. While more emphasis is shown on universal methods. The figure 2 is explaining more about the two heterodimers S100A8 and S100A9 which is fine as neonates show highly elevated blood levels of S100A8/A9 alarmins. A wholistic approach about the pathomechanism is missing.

On this point we disagree with the reviewer. The emphasis of the manuscript is on immunometabolism and its role in neonatal immunity (focus on monocytes/macrophages and carbohydrate metabolism). The figures fit very well with this focus area. 

  1. Clinical aspects and significance are missing.

We agree with the reviewer that it is important to elaborate on the clinical aspects and significance of the resent findings described in this manuscript. We included an extra paragraph in the discussion section whereby the following references were included: 95, 96 and 97 (line 469 ff.)

  1. Minor grammar issues: In line 83, “Up to know” should be written as up to now. In line 135, “immune response2” Please correct such mistakes.

We corrected those mistakes.

Round 2

Reviewer 2 Report

The authors have done commendable job to modify various aspects of the review. All the points are taken care well and improved the quality of the work. However, there are certain critical points which authors disagree to include or considered to be out of the scope of the topic. I accept the justifications for various aspects but there are few concerns which remains the same. The authors must revise few metabolic aspects which seems more repetitive as they are known phenomena and do not require rigorous details or in depth elaboration. Rather, the authors can emphasize more on the problem than explaining glycolysis, TCA etc in details. Additionally, if the same is already given in the figure, then what is the point to go in detail about these known mechanisms.

Author Response

Dear Reviewer,

thank you very much for your kind. Review. We discussed the issue regarding the "known Phenomena" with our neonatologists. While the are experts in the field of neonatal infection, they are happy to have repetition on glycolysis, since these are topics they do not deal with every day.

We thus reduced only a bit in the legenda of figure 1 and included more information on trhe regulation of GLUT3.

We hope the anuscript is now suitable for publication.

Kind regards,

Klaus Tenbrock